# Conservation Effects of Governance and Management of Sacred Natural Sites: Lessons from Vhutanda in the Vhembe Region, Limpopo Province of South Africa

**DOI:** 10.3390/ijerph19031067

**Published:** 2022-01-19

**Authors:** Ndidzulafhi Innocent Sinthumule

**Affiliations:** Department of Geography, Environmental Management and Energy Studies, University of Johannesburg, P.O. Box 524, Auckland Park, Johannesburg 2006, South Africa; isinthumule@uj.ac.za

**Keywords:** spirits, spiritual governance, management, governance, conservation, sacred natural sites, Vhutanda

## Abstract

Scholarly discourse on sacred natural sites (SNS) has focused on ecological significance, associated impacts and traditional practices as the instruments of resource governance and management. As a result, little scholarship has examined the role of spirits in governing and managing SNS; these are inhabited by deities or *numina*, commonly known as nature spirits. This study aims to provide evidence of the importance of governance by spirits as a prerequisite for protecting biophysical resources. Data were collected through semi-structured interviews and observation. The respondents were selected through a purposive sampling approach. The author also attended a funeral that was held at one of the SNS discussed in this article. The collected data were analysed through a thematic content analysis. The study shows that, for biodiversity to be protected, there should be a good relationship between humankind and the spirits. The spirits place behavioural demands on humankind involving the performance of rituals. Ritual behaviour empowers the spirits to be placed as owner of SNS and to guard against intruders. In return, humankind receives blessings, protection, patronage and governance. The governance by spirits is complemented by traditional practices. I conclude that governance by spirits should be recognised both locally and internationally.

## 1. Introduction

According to Verschuuren et al. [1] (p. 82), a sacred natural site is a “natural feature or a large area of land or water having special spiritual significance to peoples and communities”. Sacred natural sites consist of all types of natural features, including mountains, hills, streams, seeps, reefs, forests, groves, trees, rivers, lakes, lagoons, caves, islands and springs. They are of cultural or traditional significance to indigenous communities and exist as a network embedded within a territory [2,3]. They occur in various forms, including sites of ancestral or deity worship, burial grounds [4], and some are both burial sites and a place of ancestral worship [5]. The literature suggests that indigenous SNS are landscapes or “spiritscapes” that are mostly supported by the worldview based on animism and numinism [6,7]. Indigenous animism is the most ancient, geographically widespread and diverse of all belief systems and is predicated on the assumption that biophysical entities such as mountains, forests and rocks are capable of being inhabited by *numina* [8]. Studley and Jikmed use the term “spiritscapes” deliberately to denote ‘enspirited’ SNS and to differentiate them from SNS that are not inhabited by *numina* [9]. Spiritscapes in folk religion are an animistic phenomenon where landscape features, such as mountains, hills, caves, forests or bodies of water, such as lakes, rivers, pools, swamps, are inhabited by deities or *numina*, more commonly known as nature spirits [10,11,12,13]. Such spirits are associated with specific communities (e.g., custodians). The spirits are territorial or ‘cadastral’ in the sense that they are found in specific points (specific forest, cave, body of water, hill or mountain) in the landscape [3,14]. Thus, the custodians know the extent of the local spirits domain and jurisdiction [3]. The spirits’ empowerment and territorialisation does require human agency, but the spirits “call the shots” and place behavioural demands (i.e., laws) on their custodians or followers. They are the “law givers” and “law enforcers” in terms of behaviour within their jurisdiction [15]. The acknowledged owners or custodians of such sites perform ceremonies, prayer, offerings and meditation to honour and appease the spirits [8,16,17]. The rituals and ceremonies that are performed help to keep the spirits alive in SNS in return for blessings, protection, patronage and governance [18]. Thus, SNS constitute a significant part of indigenous peoples’ religion. Although they are important for religious purposes, they enjoy no legal protection and are increasingly threatened in many parts of the world due to anthropogenic pressures [19,20,21,22]. As documented by many scholars, the granting of juristic personhood has become crucial to ensure some form of legal protection [8,23,24,25]. The legal concept of juristic personhood is used for nonhuman entities (such as rivers, ecosystem, forest, mountain, etc.) when societies want to recognise them as subjects of rights and obligations [26]. SNS are not only important for religious purposes; rather, they are also globally significant because they are hotspots of biodiversity conservation [27]. Although SNS are not necessarily perceived as instruments of resource or nature conservation by the people who practice them, they nevertheless play a significant role in biodiversity conservation [19,28,29]. Institutions conserving biodiversity, including SNS, require suitable and effective governance and management approaches [30,31].

Scholars have used the term “governance” to describe a wide variety of actions and behaviours at all levels (individual, collective and institutional) of human social life and in several different settings. As a result, there is conceptual variation and little consensus on the definition of governance. Regarding nature conservation or protected areas of which SNS form a part, this paper defines governance as “the interactions among structures, processes and traditions that determine how power and responsibilities are exercised, how decisions are taken and how citizens or other stakeholders have their say” [32] (p. 2) Thus, governance is about principles, policies and rules regarding decision making. Although inextricably linked, management addresses what is performed in an area, whereas governance deals with questions of who makes decisions and how they are taken, who holds power, authority and responsibility, and how decision makers are held accountable [32,33]. Currently, the International Union for Conservation of Nature (IUCN) and the Conservation of Biological Diversity (CBD) distinguish four broad governance types: namely, governance by the government (exclusively by government agencies), shared governance (co-management, several stakeholders), private governance (exclusively privately managed), and governance by indigenous peoples and/or local communities (managed by local people) [30,34,35]. While a wealth of research has been written on governance and management of protected areas [33,36,37,38], little is known about the governance and management of SNS. A wealth of literature that has been written on SNS has focused on traditional practices or cultural codes, such as taboos and customary laws as the instruments of resource governance and management of sacred natural sites [39,40,41,42,43], despite SNS being inhabited by nature spirits. As a result, the literature has failed to recognise the cultural and spiritual significance of resident spirits in the governance and management of SNS. In addition, it has failed to recognise the cultural and ritual behaviour associated with resident spirits. Thus, there is little known about how resident spirits contribute towards the governance and management of SNS. The main research questions guiding the discussion of this paper include: How do resident spirits govern and manage SNS? How does this practice contribute to biodiversity conservation? What is the behaviour sanctioned and enjoined by the spirits and the possible forms of retribution for those who engage in non-sanctioned behaviour? What are the associated norms of customary institutions and rules that are used to complement governance by spirits in SNS in the study area? In answering these research questions, I use Vhutanda SNS in the Vhembe region, Limpopo Province of South Africa, to argue that resident spirits and the associated norms of customary institutions and rules govern and manage SNS. These practices have contributed to biodiversity conservation.

## 2. The Study Area and Notes on Methodology

### 2.1. Study Area

The study was conducted in the Vhutanda SNS, which is situated in the Thulamela local municipality under the Vhembe region in the Limpopo Province of South Africa. There are two Vhutanda SNS (Vhutanda 1, Vhutanda 2) within the Tshivhase Tea Estate (Figure 1).

This estate was established in 1973 as a joint property of the parastatal agencies, Sapekoe and Agricultural Corporation of Venda (Agriven), as part of the rural development of the Republic of Venda [44]. At that time, Venda was a Bantustan (“homeland”), which was established by the apartheid government for the Venda people (speakers of the Venda language). Both sacred sites belong to the Nevhutanda clan, who lived in the area until being displaced from their motherland by the tea estate. The sacred sites are within the village of Duthuni, which is 20 km away from Thohoyandou—the biggest town in Thulamela municipality. The villages surrounding Vhutanda SNS, and the estate include Mapate, Lwamondo and Phiphidi. At the foothill of the plantation and Vhutanda SNS lies Thathe Vondo Dam on the Mutshundudi River that originates on the southern slopes of the Soutpansberg mountain range. According to Mucina and Rutherford [45], the natural forest in the study area is classified as FOz 4 Northern Mistbelt Forest of the Zoutpansberg Mountain. The area is characterised by a species-rich evergreen afrotemperate habitat. The climate of the area comprises hot summers and cool-to-mild winters. The study area receives more than 1500 mm of rainfall per year [46,47], resulting in dense and species-diverse forest structure.

### 2.2. Data Collection Methods

Fieldwork for the study took place between April and October 2021. Interviews and observations were the main data collection methods. Semi-structured interviews were conducted with the four custodians of Vhutanda SNS, three Tshivhase Tea Estate workers, two senior Dzomo La Mupo (Voice of the Nature) officials, and five local communities. Dzomo La Mupo is a non-profit organization established in the Vhembe region with the aim of protecting and preserving the network of sacred natural sites in the area. Collectively, a total of fourteen people were interviewed. Out of the fourteen respondents who were interviewed, six were female and the remaining eight were male. Three respondents were between the ages of 30 and 50 years, and the remaining twelve interviewees were above 50 years old. Non-probabilistic purposive sampling was used to select respondents; this method was found suitable because it helped to choose information-rich cases that could provide answers to the research questions [48]. Permission to conduct the research was obtained from the local chief and the Nevhutanda family elders. In addition, Dzomo La Mupo members, local communities and Tshivhase Tea Estate workers were asked their permission before they were interviewed. All respondents were promised that their identity would not be revealed, and all data would be coded to ensure anonymity. Participants were promised that confidentiality would be maintained throughout the research process. This is in line with the eight principles of the Protection of Personal Information Act (POPIA) in South Africa.

A staff card showing my affiliation to my university helped to introduce me to the key stakeholders that were interviewed. The respondents were also informed that the study is for academic purposes and that their participation in the study would not be remunerated. They were further informed that they could withdraw their participation in this study at any time. The interview questions were designed to provide an understanding of how the resident spirits govern and manage the Vhutanda SNS and how this practice has contributed to biodiversity conservation. The questions were also aimed at comprehending the acceptable behaviour enjoined by the spirits and the possible forms of retribution (if any) for those who engage in unacceptable behaviour. The respondents were interviewed face-to-face, and all COVID-19 protocols were observed. The average duration of an interview ranged from 60 to 120 min. The interviews were conducted in Tshivenda, a local language and translated into English by a bilingual author. Although the ratio was skewed towards men (because most gatekeepers in sacred sites are male), I attempted to interview both men and women to ensure that there was representation in both groups. The author also attended a funeral that was held in one of the two sacred sites. This helped to identify the traditional processes that are followed during burials and provided an opportunity to identify some dominant species occurring within the SNS.

Observation was also used to check respondents’ behaviour and body language during interviews. The technique was used to verify whether what people said equated to what they practised [49]. Thus, observation was used to obtain ‘live’ data from naturally occurring events or social situations. While observing, pictures of different activities of interest were taken with the permission of respondents. This was important to gain a richer image of daily events rather than relying only on the information provided by interviewees.

### 2.3. Data Analysis

In line with Braun and Clarke [50], data collected through interviews were analysed using a thematic content analysis. I read all the transcribed data in the evening to familiarise myself with the data. The data were then coded; this process enabled me to organise data into manageable units and to compare data obtained through interviews [51]. From coding data, similarities (common meanings that recurred throughout the data) were identified and differences between categories were determined based on recurring patterns; results from the previous stages were also reported. The reliability and validity of the data were assessed by comparing data collected using multiple tools that included interviews and observations. In addition, events or episodes were narrated in the precise words used by the interviewees to explain to the reader the richness of the situation studied without changing the material recorded.

## 3. Results


*“… Our Zwifho (sacred natural site) are everything to us. When we are born, we are announced to our Vhadzimu (ancestral spirits) and connected to Zwifho (sacred natural site), and our spirits will return there when we die. We are alive and healthy because our Vhadzimu (ancestral spirits) protect us. In addition, our Zwifho (sacred natural site) have survived since time immemorial because they are also protected by our Vhadzimu (ancestral spirits). Without Zwifho (sacred natural site) we will perish…”*
(Interview, custodian elder, 28 April 21).

It is clear from the quotation above that Vhutanda SNS are not protected by people (security guards, soldiers and police); rather, they are protected by the ancestral spirits. This raises intriguing questions: How do the ancestral spirits protect the SNS? What are the conditions necessary for the ancestral spirits to protect the SNS? The analysis of interview materials revealed that, for ancestral spirits to protect the SNS, there must be a relationship between the spirits and humankind. The custodians of the SNS are required to perform rituals or ceremonies to pacify the spirits. In the case of the Vhutanda SNS, the custodians appease the spirits through the performance of *Thevhula* rituals. This is an annual event and the biggest sacrificial ritual or ceremony that is performed by the custodians to consult or appease the spirits. The annual performance of *Thevhula* rituals is part of maintaining good relationships with the SNS ancestral spirits. This sacrificial ritual is performed by *Makhadzi*—the senior sister of the family head or senior paternal aunt. For centuries, the *Makhadzi* has played a significant role in managing the relationships of individuals to their family, clan and community. They also are central in managing societal relationships with nature. In the case of SNS, *Makhadzi* play an important role as the mediator between the clan and the ancestral spirits. In other words, she is the one chosen to communicate with ancestral spirits on behalf of the clan. The preparation of *Thevhula* ritual starts with brewing of *Mpambo*. This is a traditional beer made from millet (*Eleusine coracana*), which is locally known as *Mufhoho*. Millet is grown locally, and each family member is required to contribute a cup of millet that is then used to create *Mpambo*.

The process of brewing *Mpambo* is perfomed by *Makhadzi*. When the *Mpambo* is ready, a traditional calabash is used to carry the brew to a chosen site at home (where the rituals are performed) or to the SNS. If this ritual is performed at home, arrows are used; the *Makhadzi* kneels down as a symbol of respect to the ancestors and pours out a libation of *Mpambo* over the arrows serving as a symbol of the departed ancestral spirits. As she pours *Mpambo*, she calls all the departed ancestors by their names. However, if the ritual is performed at the SNS, the arrows are not required because she is already with the ancestors. Here, she kneels and pours a *Mpambo* libation directly on the shrine (the designated resting place) and she communicates with the ancestors—telling the spirits what the family desires and thanking the spirits for good health and the rain. In the process of communicating with the spirits, she also gives the ancestors snuff by sprinkling it on the area where the ritual is performed. The analysis of interview materials with custodians revealed that this ritual behaviour is key to maintaining a good relationship with the ancestors. Importantly, this annual ceremony is significant in empowering the spirits to be managers, guardians or governors of SNS and to hold complete control and supremacy of all biophysical resources (plants, animals, soil) within their territory. In this sense, the power that protects the SNS does not live with the custodians, but with spirits.

Another equally important ritual that is performed at the SNS is the burial ritual. The Vhutanda clan are buried at their sacred sites, and these thus have a dual purpose. The two sites serve different purposes. Clan members who have not been circumcised, and who have never been Christians, are buried in the larger sacred site with the great grandfathers. Those who have been circumcised or who are/have been Christians are buried on the smaller sacred site; they are not buried with their ancestors as a punishment for disobeying the ancestors. In a year when there is a funeral, the Vhutanda clan do not perform *Thevhula* because the ground is ‘hot’. Rather, a burial ritual is performed at the SNS using water and snuff to connect with ancestors. This ritual is meant to ask the spirits to welcome a new member to the spirit world. In addition, if there is anything that the family desires at the time of grieving, *Makhadzi* will take the opportunity of the burial ritual to ask the spirits to grant these desires. The burial ritual also empowers the spirits and helps to maintain the potency of the SNS. By appeasing the spirits through these rituals, in return, the Vhutanda family obtain all that they desire in life, including rains, blessings, protection, governance, and good health. In addition, the ancestors can resolve family problems and conflicts. The custodians interviewed confessed that they always obtain what they ask from their ancestors, as made clear by this comment:


*“When our children and grandchildren ask for jobs and other opportunities in life, they get jobs with ease. After getting jobs, they are easily promoted to become supervisors and managers. Our ancestral spirits have never disappointed us.”*
(Interview, custodian elder, 25 October 2021).

To these people, SNS are “places to give and be given”. In this sense, performing rituals is giving the ancestors what they want and in return, the ancestors give people what they desire. Thus, the SNS are a key component of the Nevhutanda family. As a result, they cannot survive without sacred sites as made clear by the custodian member.


*“These Zwifho (sacred natural site) are the heart of our clan…We are born connected to Zwifho (sacred natural site); our blood is the blood of Zwifho (sacred natural site). So, any disorder in Zwifho (sacred natural site) will harm the Nevhutanda clan. Just as the person cannot live with their heart outside their body, neither can we live without Zwifho (sacred natural site).”*
(Interview, custodian elder, 26 October 2021).

The quotation above reveals that the custodians are connected to the SNS, and their blood is the blood of SNS. This is true because, when a child is born, she/he is introduced to the spirits. Unlike in *Thevhula* where *Mpambo* is used, in this ritual, water and snuff are used to introduce the new-born baby to the ancestors so that they can give protection to the baby. In addition, the Vhutanda clan also have Vhutanda day, which is held in March every year. This is a day of celebration near the two SNS in the presence of the ancestors. Ritual practices include the slaughtering of a cow. The study also found that ritual behaviour in Vhutanda SNS is complemented by a belief system combined with traditional methods or cultural codes, such as taboos and myths. For instance, collecting fuelwood or hunting animals inside Vhutanda SNS is taboo as made clear by a custodian elder:


*“If you cut wood in sacred natural sites they will change into a venomous snake on arrival at home. You will be shocked to find out that you have been carrying snakes. This is not a joke; it is the truth. If you don’t believe me, try it.”*
(Interview, custodian elder, 28 April 21).

It is also believed that if, one enters SNS, one will not find one’s way out. These traditional methods have thus preserved the forest, wildlife and natural resources in Vhutanda SNS allowing the area to remain pristine. Conspicuous trees that are well conserved in the Vhutanda sacred sites include tropical wild quince (*Cryplocarya lieberliana*), ironwood (*Drypeles gerrardii*), forest peach (*Rawsonia lucida*), bushwillow (*Combrelum kraussii*), natal fig (*Ficus natalensis*), yellowwood (*Podocarpus lalifolius*), lemonwood (*Xymalos monospora*), cabbage tree (*Cussonia spicala*), and whipstick Falseloquat (*Oxyanthus gerrardii*). Although these species are abundant at the sacred site, they have been depleted in the surrounding areas.

The analysis of interview materials with the custodians revealed that there are behaviours that are acceptable and those that are not acceptable by the spirits in Vhutanda SNS. While the performing of rituals in SNS is acceptable, behaviours including cutting of wood, trespassing and hunting are unacceptable. The latter behaviours in the Vhutanda SNS lead to retribution by spirits. For instance, in the 1950s, Caucasian people camped near Vhutanda SNS intending to hunt animals in the area. Unfortunately, they accidentally entered the SNS and were held hostage by the ancestral spirits (they could not find their way out of the forest) until the elders of the Nevhutanda family were asked to request the spirits to release them. To ensure that they did not find themselves inside the sacred sites, the Caucasian people constructed a fence that was uprooted overnight by the ancestral spirits (guardian spirits) (Interview, custodian elder, 24 October 2021). Similarly, in the late 1960s, a Caucasian officer on a tractor or bulldozer who stubbornly ignored warnings against clearing the Vhutanda sacred site mysteriously disappeared in the sacred sites as he attempted to clear it (Interviews, custodian elder, 24 October 2021; Community member, 25 October 2021). This explains why the Vhutanda SNS remain within the tea plantation. In addition, in 2010, an aircraft that was conducting aerial spraying of fertiliser onto the tea crop at Tshivhase tea estate allegedly passed above the Vhutanda sacred site. It crashed immediately and was destroyed beyond repair; it is alleged that the noise it was making disrupted the ancestral spirits (Interview, custodian elder, 25 October 2021). After the incident, the baboons from Vhutanda SNS visited the area where the incident happened and created a circle around the aircraft debris, after which they went back to the sacred site (Interview, Tshivhase Tea Estate worker, 24 October 2021).

## 4. Discussion

The main finding emerging from this study is that there needs to be a good relationship between humankind and the ancestors. Humankind has the responsibility of performing rituals and ceremonies regularly to honour and appease the spirit world. In the case of the study area, rituals (Thevhula) are performed on an annual basis or when there are funerals. The ritual behaviours empower the spirits not only to be owners, but also ‘governors’ of the sites; the spirits hold “total power and authority” over all the biophysical resources within their domain [52,53]. The *numina* decide on the objectives of governance and their pursuance and orchestrate them through intermediaries in the decision-making process [8]. Thus, behaviour in SNS inhabited by spirits is not determined by the Western concept of “governance”, but rather by daily rituals and annual ceremonies [9] in order to honour and appease the spirits world or other-than-human world [54]. The term “other-than-human” refers to a conceptual shift in anthropology and other social sciences seeking to avoid human exceptionalism and, instead, extending the social to other entities (whether they are ancestors, spirits, etc.) [55]. The other-than-human world is part of the “ontological turn” exemplified in the posthuman [55], “more-than-human” [56], pluriversal [57,58] and “a world of many worlds” [59]. Posthuman or other-than-human-centred approach does not imply the separation of humans, non-humans, and more-than humans into independent domains; rather, it is about their interdependencies or relationships [60], which is achieved through rituals and liturgies. Ritual behaviour in SNS is part of maintaining good relations with the other-than-human world, which is translated as the maintenance of the “topocosmic equilibrium” as in the case of the Tibetan sacred mountains in southwest China [9,61]. “Topocosmic equilibrium” is a translation of the Tibetan “*snod bcud do mnyam*” and expands on the Gaster term “topocosm”, a word for the entire locality including all biophysical resources [62]. Topocosmic equilibrium is the customary ritual maintenance of harmony and good relations between all the elements of the cosmos [9]. This is often part of “contractual relations of reciprocity” between the spirits and humankind [63]. The “contractual relations of reciprocity” is just one example of the “legal relationship” that humans have with spirits [64] in which humankind does not hold a place of natural authority. In this sense, the spirits in Vhutanda SNS place ritual behavioural demands on custodians or humankind, and in return, humankind receives blessings, protection, rain, good health, governance and anything that they may desire. Studley and Horsley [18] termed this form of governance and its associated cultural and ritual behavioural practice “spiritual governance”. The term “spiritual governance” was adopted by Studley [8] to describe the “governance” by *numina* of SNS on the Tibetan Plateau, and he later discovered its use by Bellezza [17]. It has gained some traction within the literature [1,65]. Although this type of governance is significant and has contributed to biodiversity conservation in SNS, it is not recognised by either the IUCN or the CBD. However, if we are to protect the biodiversity of the world (in formal and informal protected areas), there is compelling justification for expanding the concept of governance so that it also embraces spiritual governance.

The study also found that, although the spirits are dependent upon humankind as their main actors, they do not consult with humankind when making decisions. Thus, all decisions in terms of what constitutes acceptable behaviour in SNS is determined by spirits without consulting humankind. In this sense, as Studley and Horsley [18] have noted, the spirits are autocratic in terms of governance and are not in line with the principles of good governance (legitimacy and voice, direction, performance, accountability, fairness and rights) as outlined by Graham et al. [32]. Ever since the works of Jeremy Bentham (1748–1832), both law and governance have become godless, secularised and democratic [66]. In contrast, the spiritual dimension provides the basis for autochthonous law and governance [67,68], as it once did for Blackstonian law [69]. Blackstone’s understanding of law was firmly rooted in God’s revelation in nature and in Scripture. All human activity is governed by law, and every person is answerable to law, which is revealed expressly by God [70]. From an indigenous animistic perspective, Graham’s definition of governance is not “good” and does not make much sense because spiritual autocracy may also be considered “good” governance. Although the aim of spiritual governance is not “biodiversity conservation” (which is often negatively viewed by indigenous people as a Western scientific intervention), the behavioural practices of local people (enjoined by the spirit) result in the ritual protection/nurture of flora and fauna.

The Vhutanda SNS are “no-go areas” and remain the most powerful and respected sacred sites not only in the Vhembe region, but in the southern African region. Despite SNS being affected by human activities and therefore losing value in other parts of the world [19,22,43,71], the Vhutanda SNS remain pristine. Sharma et al. [72] also found that in Central Himalaya, the prohibition of resource harvest from SNS has also contributed to dense reservoirs of highly valued trees providing seed stock for further regeneration. Similarly, the enhancement of biodiversity and the conservation of flora and fauna can also be seen in the deity-inhabited SNS of Bhutan [73]. Although the institution of spiritual governance is not necessarily perceived as an instrument of resource or nature conservation by the people who practice them, they nevertheless play a significant role in biodiversity conservation [19]. Although Vhutanda SNS is powerful, respected, significant for religious purposes, and serve as a hotspot of biodiversity, it enjoys no legal protection. Thus, the custodians do not have property rights or legal and intellectual ownership over their SNS [74]. The concept of juristic personhood has become important when dealing with protection of biological entities or natural resources (mountains, rivers, forests, etc.). Central to juristic personhood is the view that nature has rights [75,76]. Personhood means that lawsuits to protect a particular natural resource or a biological entity can be brought on behalf of that natural resource [15]. The granting of juristic personhood to SNS is crucial to ensure some form of legal protection that will complement the community-based customary ritual protection that is already in place. For instance, juristic personhood has been used successfully for purposes of litigation in Ecuador where the plaintiff was a river [15,76]. Similarly, ‘nature rights’ or biophysical entities have also been recognised in New Zealand, Colombia, Canada, India, and the USA [8].

The study also found that spiritual governance is supplemented by belief systems and traditional practices or cultural codes, such as taboos and myths. Traditional practices represent a class of informal institutions, where cultural norms, rather than governmental judicial laws and rules, define human behaviour and conduct towards the natural environment [77,78], whereas belief systems hold that certain landscapes are inhabited by spirits and any violation of cultural codes, such as taboos, will lead to retribution [42]. For instance, angry spirits may cause sickness, calamity, crop failure, and disaster upon custodians and the whole community, as is the case in the Tibetan [18]. As documented by scholars, the traditional methods and belief systems of communities are a powerful force driving the conservation of SNS in many parts of the world [42,79,80]. Unlike in other studies where traditional methods and belief systems are seen as the only instruments for protecting biophysical resources in SNS [39,41,43], in the study area, the belief system and traditional practices are incorporated into daily living so that compliance with them ensures adherence to the governance by spirits. Thus, traditional practices complement spiritual governance.

While there are sanctioned ritual behaviours that are acceptable or enjoined by the spirits, non-sanctioned behaviours, such as trespassing, cutting of wood, hunting in SNS or any other activities that disturb or anger the spirits, lead to retribution. This can be in the form of disease, natural disaster, or other misfortunes befalling the offenders. Retribution or supernatural punishment for non-sanctioned behaviour (such as trespassing, cutting of wood, hunting) in the SNS is also common in many parts of the world [8,63,81,82,83]. The honouring and appeasing of angry spirits by rectifying the harm conducted to them and the habitats they rule over remains central to the relationship between indigenous communities and spirits [17].

## 5. Conclusions

This study has shown that, for SNS to exist, there should be a good relationship between humankind and the spirits. Humankind is responsible for honouring and appeasing the spirits through the regular performance of rituals (Thevhula). The rituals empower the spirits to remain alive in SNS; these spirits are therefore responsible for governing and managing SNS. Spiritual governance is complemented by belief systems and traditional practices or cultural codes. Any violation of traditional methods or cultural codes, such as taboos, leads to retribution by the spirits in SNS. Such retribution can be in the form of disease, natural disaster, or other misfortunes to befall offenders. It can be concluded that spiritual governance and its associated cultural and ritual behaviour has contributed to the conservation of biodiversity in the study area. The results of biodiversity conservation offered by formally protected areas are similar to the results offered by SNS. As a result, spiritual governance should not only be embraced by conservationists, but also by local and national governments in other areas where they have SNS that are inhabited by numina. Thus, such governance by spirits should not only be infused into the constitution, but also into existing environmental rules and regulations governing nature in the Republic of South Africa, Africa and other continents that have SNS. The recognition of governance by spirits by various states all over the world will set a stage for this kind of governance to also be recognised by the IUCN and CBD. Essentially, the IUCN and CBD governance matrix that only recognises four governance types (namely, state governance, private governance, co-management and community governance) will require updating to include governance by spirits. In an era where most sacred sites are under threat from anthropogenic activities, this study recommend that efforts should be made to ensure that SNS are granted juristic personhood or natural rights to complement community-based customary ritual protection, belief systems and traditional practices that is already in place. The granting of juristic personhood is crucial and will ensure some form of legal protection to SNS (and resident *numina*). The recognition of SNS as juristic persons not only in South Africa, but also in other parts of the world could lead to their recognition by the IUCN, which will be a benefit not only for nature conservation, but also for indigenous people.

## Figures and Tables

**Figure 1 ijerph-19-01067-f001:**
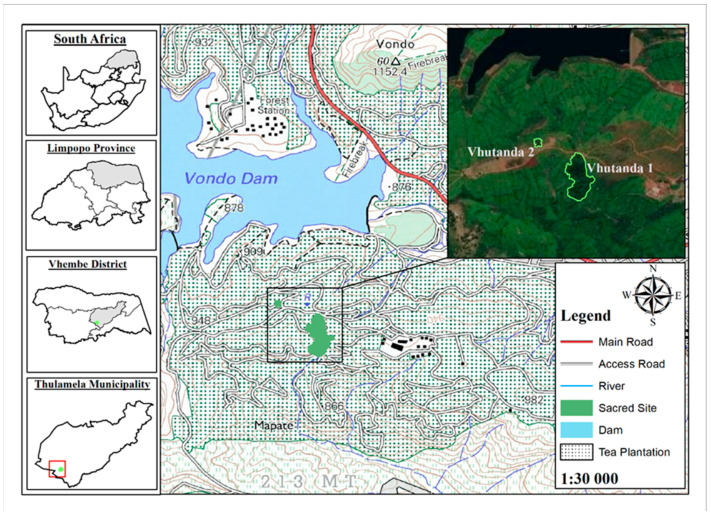
Location of Vhutanda SNS in the Vhembe region, Limpopo Province of South Africa.

## Data Availability

The data presented in this study are available on request from the corresponding author.

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
