# Peer review of "Conservation Effects of Governance and Management of Sacred Natural Sites: Lessons from Vhutanda in the Vhembe Region, Limpopo Province of South Africa"

_ijerph, 2022, doi:10.3390/ijerph19031067_

Round 1

Reviewer 1 Report

Limpopo

A well written cohesive article that embraces most of the concepts that underpin the literature on Sacred Natural Sites. The author, however, does not always

  • source or discuss some concepts, namely the other-than-human world and juristic personhood. The former should be discussed in the context of the posthuman “ontological turn” and the latter in the context of the legal protection of Sacred Natural Sites
  • reference the most current material on Sacred Natural Sites, Spiritual Governance, Juristic Personhood, and the posthuman (other-than-human)

Furthermore on one occasion the author refers to the review of a book rather than the book itself.

A partial re-write is suggested.

Comments on Text

Line 27 Can you use more up to date references for SNS. Why not consider Verschuuren et al 2021 which makes reference to Sacred Natural Sites, spiritual governance, juristic personhood and the IUCN governance matrix.

Line 32 Studley and Jikmed Awang use the term “spiritscapes” deliberately to denote ‘enspirited’ Sacred Natural Sites and to differentiate them from Sacred Natural Sites that are not inhabited by numina.

Line 33 How are you defining animism? Whose definition are you using? How do Dudley et al 2005 or UNESCO 2003 define animism? Indigenous animism (See Studley 2019:14) does differ from both Tylor (1871) and Harveys (2005) definition of animism.

Line 33 It might make more sense when coupled with animism to use numinism. The origin of the term animistic numina (called “numinism”) originates mainly from a quotation by Augustine of Hippo citing information derived from Varro (Johnston et al 2016:27)

Line 38 Can you say more on territorial or cadastral spirits and geospatial boundaries to their territory. Do the people of Vhutanda know the extent of the local spirits domain and jurisdiction?

Line 39 I am not sure what you mean when you say “The spirits are ...’regulated verbally’ rather than by a written law”. The spirits empowerment and territorialisation does require human agency but the spirits “call the shots” and place behavioural demands (i.e. laws) on their followers. They are the “law givers” and “law enforcers” in terms of behaviour within their jurisdiction (See Studley and Bleisch 2018). You might want to mention by way of example that in ancient Greek society divine law (themis) was differentiated from mans law (nomos). Furthermore within all parts of Aboriginal society, the laws and rules of behaviour were set in the dreamtime by the ancestral creation beings that are said to have given the people the laws they were to live by https://austhrutime.com/law_and_order.htm.

Line 40 I am not sure why you mention “built shrines” in the context of Sacred Natural Sites. I do recognise that in some cases SNS are demarcated, typically (in High Asia) with piles of stones. Furthermore the spirits are rarely iconographically represented.

Line 42 In most animistic societies local people seek to both honour and appease the spirits? Perhaps this is not the case in Limpopo Province?

Line 42 I am slightly puzzled why a review of Studley’s book (2019) by Verschuuren (2019) is referenced rather than the book itself given that the book is very apposite.

Line 274 Do you need to add honour to appease?

Line 278 Do you need to add more to “total power and authority”. The numina decide on the objectives of governance and their pursuance and orchestrate them through intermediaries in the decision-making process. (See Studley 2019)

Line 282 I think you mean “other-than-human” world and you need to source this (Hallowell 2002:21). You might want to discuss “other-than-human” worlds under the aegis of the “ontological turn” exemplified in the posthuman (Blanco-Wells 2021) “more-than-human” (Blaser 2010), the pluriversal (Kothari et al 2018 , Escobar 2017) and “worlds within worlds” (Stahler-Sholk 2000) which are being collectively reworked by indigenous people for conservation purposes (de la Cadena 2010)

Line 283 “topocosmic equilibrium” is a translation of the Tibetan snod bcud do mnyam and expands on Gaster term “topocosm” a word for the entire locality--soils, plants, animals, everything, yesterday, today, and tomorrow--taken as an organism (Gaster 1961)

Line 285 I think you mean “of the cosmos”

Line 286 The “contractual relations of reciprocity” is just one example of the “legal relationship” that humans have with spirits (See Petrazycki 2011:192) in which humankind does not hold a place of natural authority.

Line 291 The term “spiritual governance” was adopted by Studley (2019) to describe the “governance” by numina of Sacred Natural Sites on the Tibetan Plateau, and he later discovered its use by Bellezza (2014). It has gained some traction within the literature (Meir 2021, Verschuuren et al 2021).

Line 299 While it is of academic interest to contrast autocratic governance with Graham’s definition (2003) spiritual autocracy may also be considered “good” governance from an indigenous perspective. I think it’s important to take the contrast further. Ever since Bentham (1748-1832) both law and governance have become godless, secularised and democratic (2018). In contrast the spiritual dimension provides the basis for autochthonous law and governance (Glenn 2000, Christensen 2021) as it once did for Blackstonian law (1966). Blackstone's understanding of law was firmly rooted in God's revelation in nature and in Scripture. All human activity is governed by law, and every person is answerable to law, which is revealed expressly by God (Wilsey 2011). From an indigenous animistic perspective Graham’s definition of governance is not “good” and does not make much sense.

Line 303 Perhaps it is important to point out that the aim of spiritual governance is not “biodiversity conservation” (which is often negatively viewed by indigenous people as a western scientific intervention) but the behavioural practices of local people (enjoined by the spirit) result in the ritual protection/nurture of flora and fauna.

Line 319 Perhaps you could say more on the forms of retribution that take place when local people engage in non-sanctioned behaviour (See Studley 2019)

Line 331 Perhaps you could say more on the forms of non-sanctioned behaviour (See Studley 2019)

Line 332 Should that be honouring and appeasing?

Line 332 Should that be honouring and appeasing?

Line 354 I think you should make reference to juristic personhood much earlier (at least in the discussion and preferably in the literature review) and you do need to reference it (Alley 2019, O’Donnell 2019, Smith 2021, Studley 2019). The granting of juristic personhood to Sacred Natural Sites (and resident numina) is crucial to insure some form of legal protection and it has been used successfully for purposes of litigation in Ecuador where the plaintiff was a river,

References

Alley, K. (2019) ‘River Goddesses, Personhood and Rights of Nature: Implications for Spiritual Ecology’. Religions 10 (9 502)

Bentham, J. (2018) An Introduction to the Principles of Morals and Legislation. CreateSpace Independent Publishing Platform

Blackstone, W. (1966) Commentaries on the Laws of England. London: Dawsons of Pall Mall

Blanco-Wells, G. (2021) ‘Ecologies of Repair: A Post-Human Approach to Other-Than-Human Natures’. Frontiers in Psychology 12:633737

Blaser, M. (2010) Storytelling Globalization from the Chaco and Beyond Duke University Press. available from

Christensen, G. (2021) ‘Indigenous Perspectives on Corporate Governance, 23’. Journal of Business Law 902

De la Cadena, M. (2010) ‘Indigenous Cosmopolitics in the Andes: Conceptual Reflections beyond “Politics”’. Cultural Anthropology 25 (2), 334–370

Escobar, A. (2017) ‘Sustaining the Pluriverse:The Political Ontology of Territorial Struggles in Latin America’. in The Anthropology of Sustainability: Beyond Development and Progress. ed. by Brightman, M. and Lewis, J. Palgrave Macmillan, 237–26

Gaster, T. (1961) Thespis: Ritual, Myth and Drama in the Ancient Near East. New York: Anchor

Glenn, P. (2000) Legal Traditions of the World. Oxford University Press

Hallowell, A.I. (2002) ‘Ojibwa Ontology, Behavior, and World View’. in Readings in Indigenous Religions ed. by Harvey, G. vol. 22. London & New York: Bloomsbury Publishing, 17–49.

Harvey, G. (2005) Animism: Respecting the Living World. New York: Columbia University Press

Johnston, P., Mastrocinque, A., and Papaioannou, S. (2016) Animals in Greek and Roman Religion. Cambridge Scholars Publishing

Kothari, A., Salleh, A., Escobar, A., Acosta, A., and Demaria, F. (2019) Pluriverse: A Post-Development Dictionary. Tulika Books

Meir, D. (2021) ‘The Gzhi Bdag Cult as a Regenerative Worldview and an Animistic Expression of Biocultural Resistance in the Hengduan Mountain Range’. The Tibet Journal XLVI (1), 25–75

O’Donnell, E. (2019) Legal Rights for Rivers: Competition, Collaboration, and Water Governance. Routledge

Smith, J. (2021) ‘Nature Is Becoming a Person’. Foreign Policy (Nov 24).

Stahler-Sholk, R. (2000) ‘A World in Which Many Worlds Fit: Zapatista Responses to Globalization’. Latin American Studies Association, Miami, Florida.

Studley, J. (2019) Indigenous Sacred Natural Sites and Spiritual Governance: The Legal Case for Juristic Personhood. Routledge

Studley, J. and Bleisch, B. (2018) ‘Juristic Personhood for Sacred Natural Sites: A Potential Means for Protecting Nature’. Parks 24 (1), 81–96

Tylor, E. (1871) Primitive Culture. London: John Murray

Verschuuren, B., Mallarach, J.-M., Bernbaum, E., Spoon, J., Brown, S., Borde, R., Brown, J., Calamia, M., Mitchell, N.J., Infield, M., Lee, E., and Groves, C. (2021) Cultural and Spiritual Significance of Nature : Guidance for Protected and Conserved Area Governance and Management. ed. by Groves, C. Best Practice Protected Area Guidelines Series. vol. 32. Gland, Switzerland: IUCN

Wilsey, J. (2011) One Nation Under God? An Evangelical Critique of Christian America. Wipf and Stock Publishers

Author Response

Reviewer 1

Dear Reviewer 1

Thanks for your useful comments that has helped me to improve my manuscript. Details of how I have addressed your comments appear on the table below.

Comments

Response

A well written cohesive article that embraces most of the concepts that underpin the literature on Sacred Natural Sites. The author, however, does not always source or discuss some concepts, namely the other-than-human world and juristic personhood. The former should be discussed in the context of the posthuman “ontological turn” and the latter in the context of the legal protection of Sacred Natural Sites reference the most current material on Sacred Natural Sites, Spiritual Governance, Juristic Personhood, and the posthuman (other-than-human)

This useful comment has been addressed. The other-than-human world has been discussed in the context of the posthuman “ontological turn” as suggested on page 10. The concept of juristic personhood has been discussed on page 2 under the section on literature review and on page 12 under the section on discussion.

Furthermore on one occasion the author refers to the review of a book rather than the book itself.

The book review reference has been removed and replaced with the book itself on page 2.

Comments on Text

Line 27 Can you use more up to date references for SNS. Why not consider Verschuuren et al 2021 which makes reference to Sacred Natural Sites, spiritual governance, juristic personhood and the IUCN governance matrix.

The definition and reference has been updated on page 1 and I have now used Verschuuren et al 2021’s definition as suggested by the reviewer.

Line 32 Studley and Jikmed Awang use the term “spiritscapes” deliberately to denote ‘enspirited’ Sacred Natural Sites and to differentiate them from Sacred Natural Sites that are not inhabited by numina.

This important line have been added on page 2.

Line 33 How are you defining animism? Whose definition are you using? How do Dudley et al 2005 or UNESCO 2003 define animism? Indigenous animism (See Studley 2019:14) does differ from both Tylor (1871) and Harveys (2005) definition of animism.

In line with Studley 2019, a definition of indigenous animism has been given on page 1.

Line 33 It might make more sense when coupled with animism to use numinism. The origin of the term animistic numina (called “numinism”) originates mainly from a quotation by Augustine of Hippo citing information derived from Varro (Johnston et al 2016:27)

This has been corrected on page 1.

Line 38 Can you say more on territorial or cadastral spirits and geospatial boundaries to their territory.

This has been clarified on page 2.

Line 39 I am not sure what you mean when you say “The spirits are ...’regulated verbally’ rather than by a written law”. The spirits empowerment and territorialisation does require human agency but the spirits “call the shots” and place behavioural demands (i.e. laws) on their followers. They are the “law givers” and “law enforcers” in terms of behaviour within their jurisdiction (See Studley and Bleisch 2018). You might want to mention by way of example that in ancient Greek society divine law (themis) was differentiated from mans law (nomos). Furthermore within all parts of Aboriginal society, the laws and rules of behaviour were set in the dreamtime by the ancestral creation beings that are said to have given the people the laws they were to live by https://austhrutime.com/law_and_order.htm.

This has been corrected and clarified on page 2.

Line 40 I am not sure why you mention “built shrines” in the context of Sacred Natural Sites. I do recognise that in some cases SNS are demarcated, typically (in High Asia) with piles of stones. Furthermore the spirits are rarely iconographically represented.

This important comment has been corrected on page 2.

Line 42 In most animistic societies local people seek to both honour and appease the spirits? Perhaps this is not the case in Limpopo Province?

The word honour has been addedd on page 2.

Line 42 I am slightly puzzled why a review of Studley’s book (2019) by Verschuuren (2019) is referenced rather than the book itself given that the book is very apposite.

The book itself by Studley has been referenced.

Line 274 Do you need to add honour to appease?

The word honour has been added on page 10.

Line 278 Do you need to add more to “total power and authority”. The numina decide on the objectives of governance and their pursuance and orchestrate them through intermediaries in the decision-making process. (See Studley 2019)

In line with Studley 2019, the line The numina decide on the objectives of governance and their pursuance and orchestrate them through intermediaries in the decision-making process has been added on page 10.

Line 282 I think you mean “other-than-human” world and you need to source this (Hallowell 2002:21). You might want to discuss “other-than-human” worlds under the aegis of the “ontological turn” exemplified in the posthuman (Blanco-Wells 2021) “more-than-human” (Blaser 2010), the pluriversal (Kothari et al 2018 , Escobar 2017) and “worlds within worlds” (Stahler-Sholk 2000) which are being collectively reworked by indigenous people for conservation purposes (de la Cadena 2010)

This has been corrected on page 10 and the concept of other-than-human world has been discussed under the aegis of the “ontological turn” on page 10.

Line 283 “topocosmic equilibrium” is a translation of the Tibetan snod bcud do mnyam and expands on Gaster term “topocosm” a word for the entire locality--soils, plants, animals, everything, yesterday, today, and tomorrow--taken as an organism (Gaster 1961)

This has been added on page 11.

Line 285 I think you mean “of the cosmos”

This has been corrected on page 11.

Line 286 The “contractual relations of reciprocity” is just one example of the “legal relationship” that humans have with spirits (See Petrazycki 2011:192) in which humankind does not hold a place of natural authority.

This line has been added on page 11.

Line 291 The term “spiritual governance” was adopted by Studley (2019) to describe the “governance” by numina of Sacred Natural Sites on the Tibetan Plateau, and he later discovered its use by Bellezza (2014). It has gained some traction within the literature (Meir 2021, Verschuuren et al 2021).

This has been corrected on page 11.

Line 299 While it is of academic interest to contrast autocratic governance with Graham’s definition (2003) spiritual autocracy may also be considered “good” governance from an indigenous perspective. I think it’s important to take the contrast further. Ever since Bentham (1748-1832) both law and governance have become godless, secularised and democratic (2018). In contrast the spiritual dimension provides the basis for autochthonous law and governance (Glenn 2000, Christensen 2021) as it once did for Blackstonian law (1966). Blackstone's understanding of law was firmly rooted in God's revelation in nature and in Scripture. All human activity is governed by law, and every person is answerable to law, which is revealed expressly by God (Wilsey 2011). From an indigenous animistic perspective Graham’s definition of governance is not “good” and does not make much sense.

A further contrast on spiritual autocratic governance has been given on page 11.

Line 303 Perhaps it is important to point out that the aim of spiritual governance is not “biodiversity conservation” (which is often negatively viewed by indigenous people as a western scientific intervention) but the behavioural practices of local people (enjoined by the spirit) result in the ritual protection/nurture of flora and fauna.

This important comment has been addressed on page 12.

Line 319 Perhaps you could say more on the forms of retribution that take place when local people engage in non-sanctioned behaviour

The forms of retribution that take place when local people engage in non-sanctioned behaviour is given on page 13.

Line 331 Perhaps you could say more on the forms of non-sanctioned behaviour.

Some example on non-sanctioned behaviour in sacred natural sites has been given on page 13.

Line 332 Should that be honouring and appeasing?

This has been corrected on page 13.

Line 354 I think you should make reference to juristic personhood much earlier (at least in the discussion and preferably in the literature review) and you do need to reference it (Alley 2019, O’Donnell 2019, Smith 2021, Studley 2019). The granting of juristic personhood to Sacred Natural Sites (and resident numina) is crucial to insure some form of legal protection and it has been used successfully for purposes of litigation in Ecuador where the plaintiff was a river,

This important comment has been addressed by making reference to juristic personhood much earlier on page 2 and 12.

References

Alley, K. (2019) ‘River Goddesses, Personhood and Rights of Nature: Implications for Spiritual Ecology’. Religions 10 (9 502)

Bentham, J. (2018) An Introduction to the Principles of Morals and Legislation. CreateSpace Independent Publishing Platform

Blackstone, W. (1966) Commentaries on the Laws of England. London: Dawsons of Pall Mall

Blanco-Wells, G. (2021) ‘Ecologies of Repair: A Post-Human Approach to Other-Than-Human Natures’. Frontiers in Psychology 12:633737

Blaser, M. (2010) Storytelling Globalization from the Chaco and Beyond Duke University Press. available from

Christensen, G. (2021) ‘Indigenous Perspectives on Corporate Governance, 23’. Journal of Business Law 902

De la Cadena, M. (2010) ‘Indigenous Cosmopolitics in the Andes: Conceptual Reflections beyond “Politics”’. Cultural Anthropology 25 (2), 334–370

Escobar, A. (2017) ‘Sustaining the Pluriverse:The Political Ontology of Territorial Struggles in Latin America’. in The Anthropology of Sustainability: Beyond Development and Progress. ed. by Brightman, M. and Lewis, J. Palgrave Macmillan, 237–26

Gaster, T. (1961) Thespis: Ritual, Myth and Drama in the Ancient Near East. New York: Anchor

Glenn, P. (2000) Legal Traditions of the World. Oxford University Press

Hallowell, A.I. (2002) ‘Ojibwa Ontology, Behavior, and World View’. in Readings in Indigenous Religions ed. by Harvey, G. vol. 22. London & New York: Bloomsbury Publishing, 17–49.

Harvey, G. (2005) Animism: Respecting the Living World. New York: Columbia University Press

Johnston, P., Mastrocinque, A., and Papaioannou, S. (2016) Animals in Greek and Roman Religion. Cambridge Scholars Publishing

Kothari, A., Salleh, A., Escobar, A., Acosta, A., and Demaria, F. (2019) Pluriverse: A Post-Development Dictionary. Tulika Books

Meir, D. (2021) ‘The Gzhi Bdag Cult as a Regenerative Worldview and an Animistic Expression of Biocultural Resistance in the Hengduan Mountain Range’. The Tibet Journal XLVI (1), 25–75

O’Donnell, E. (2019) Legal Rights for Rivers: Competition, Collaboration, and Water Governance. Routledge

Smith, J. (2021) ‘Nature Is Becoming a Person’. Foreign Policy (Nov 24).

Stahler-Sholk, R. (2000) ‘A World in Which Many Worlds Fit: Zapatista Responses to Globalization’. Latin American Studies Association, Miami, Florida.

Studley, J. (2019) Indigenous Sacred Natural Sites and Spiritual Governance: The Legal Case for Juristic Personhood. Routledge

Studley, J. and Bleisch, B. (2018) ‘Juristic Personhood for Sacred Natural Sites: A Potential Means for Protecting Nature’. Parks 24 (1), 81–96

Tylor, E. (1871) Primitive Culture. London: John Murray

Verschuuren, B., Mallarach, J.-M., Bernbaum, E., Spoon, J., Brown, S., Borde, R., Brown, J., Calamia, M., Mitchell, N.J., Infield, M., Lee, E., and Groves, C. (2021) Cultural and Spiritual Significance of Nature : Guidance for Protected and Conserved Area Governance and Management. ed. by Groves, C. Best Practice Protected Area Guidelines Series. vol. 32. Gland, Switzerland: IUCN

Wilsey, J. (2011) One Nation Under God? An Evangelical Critique of Christian America. Wipf and Stock Publishers

Suggested references have been used throughout the paper.

Reviewer 2 Report

The submitted submission is an interesting and professionally planned text addressing an interesting issue related to spiritual approaches to territory management and the functioning of local societies.
Particularly noteworthy is the detailed analysis of the theoretical basis of the issue and the literature review. It should be noted that the author has made a very detailed presentation of the research methodology and logically explained its relevance to the planned observations and research.

However, such a detailed description of the methodology was not followed by presentations of the results of the observations, which the author reduces to some general descriptions of the rituals (which could have been done equally well on the basis of descriptions from the literature). There was no specification of the characteristics of the research sample (how many people were surveyed, their age, their social position, etc.) or even a basic statistical treatment of the survey results. 
It would be useful to add some original insights from the interviewees or interviewees regarding the way the territory is managed through spirits, the importance of the natural landscape in preserving the spiritual heritage of the society, etc.

In conclusion - it would be useful to clearly indicate the conclusions resulting from the research conducted, to propose the application of existing experiences in this field or the planning of new ones, and to indicate the possibility of applying the observations in other regions with a similar sensitivity and socio-spiritual traditions.

Author Response

Reviewer 2

Dear Reviewer 2

Thanks for your useful comments that has helped me to improve my manuscript. Details of how I have addressed your comments appear on the table below.

Comments

Response

The submitted submission is an interesting and professionally planned text addressing an interesting issue related to spiritual approaches to territory management and the functioning of local societies. Particularly noteworthy is the detailed analysis of the theoretical basis of the issue and the literature review. It should be noted that the author has made a very detailed presentation of the research methodology and logically explained its relevance to the planned observations and research. However, such a detailed description of the methodology was not followed by presentations of the results of the observations, which the author reduces to some general descriptions of the rituals (which could have been done equally well on the basis of descriptions from the literature).

Thanks for this useful comment. This study focuses on spiritual governance of sacred natural site. The emphasis of spiritual governance is on the performance of rituals by the custodians. Thus, the relationship between the custodians and the spirits through performance of the rituals is the core of this research project (spiritual governance) and this information has been thoroughly explained from pages 6-10. The discussion given on page 6-10 is not only general description of rituals but how the performance of rituals by the custodians empower the spirits to remain alive in sacred natural sites and how these spirits are therefore responsible for governing and managing sacred natural sites.

There was no specification of the characteristics of the research sample (how many people were surveyed, their age, their social position, etc.) or even a basic statistical treatment of the survey results.

The characteristics of the research sample (how many people were surveyed, their age, their social position, etc.) has now been given on page 5.

With regard to basic statistics, no statistical analysis were performed in this study. Rather, the data were analysed through thematic content analysis which is explained in page 6.

It would be useful to add some original insights from the interviewees or interviewees regarding the way the territory is managed through spirits, the importance of the natural landscape in preserving the spiritual heritage of the society, etc.

A detailed description on how the territory is managed by spirits and the importance of such landscapes in preserving bio-cultural heritage is explained from pages 6-10. In addition, events or episodes (original insights from interviewees) were narrated in the precise words used by the interviewees to explain to the reader the richness of the situation studied without changing the material recorded.

In conclusion - it would be useful to clearly indicate the conclusions resulting from the research conducted, to propose the application of existing experiences in this field or the planning of new ones, and to indicate the possibility of applying the observations in other regions with a similar sensitivity and socio-spiritual traditions.

The conclusion resulting from the study conducted is provided and the possibility of applying the observations in other regions has also been given on page 14.

Round 2

Reviewer 2 Report

The author has clarified all doubts and made the necessary corrections which have made the text more readable. My doubts have also been dispelled by a substantive explanation of the changes applied, as well as the justification for not making such changes.